# Predictors of radiological aggravations of pulmonary MAC disease

**Norio Kodaka, Chihiro Nakano, Takeshi Oshio, Kayo Watanabe, Kumiko Niitsuma, Chisato Imaizumi, Hiroto Matsuse**  *

Division of Respiratory Medicine, Department of Internal Medicine, Toho University Ohashi Medical Center, Tokyo, Japan

* hiroto.matsuse@med.toho-u.ac.jp

**Data Availability Statement:** Data contain potentially identifying information and the Ethics Committee of Toho University Ohashi Medical Center has imposed restrictions on making the data publicly available. Requests for data may be

## Abstract

### Background and objectives

The number of patients with pulmonary *Mycobacterium avium* complex (MAC) disease is increasing worldwide, especially among middle-aged women and never-smokers.

However, little is known about the factors causing exacerbations of pulmonary MAC disease in untreated patients. The aim of the present study was to identify the predictors of radiological aggravations of pulmonary MAC disease.

### Methods

From April 2011 to December 2018, 238 MAC patients at our institute were newly diagnosed with pulmonary MAC disease according to the 2007 American Thoracic Society/Infectious Disease Society guideline. Their medical records were examined retrospectively for their clinical findings. The radiological findings at the time of the diagnosis and 1 year later were evaluated. To identify the predictors of radiological aggravation, multivariable analysis was performed with the data of 167 treatment-naïve patients.

### Results

Female, never-smoker, and nodular/bronchiectatic (NB) type were predominant in patients with pulmonary MAC disease. Univariate analysis of data from treatment-naïve subjects showed that no lung diseases other than MAC, extensive radiological findings, and a positive acid-fast bacilli (AFB) smear were significantly associated with radiological aggravations. On multivariate analysis, the radiological factor (larger affected area) and absence of other lung disease were significantly associated with radiological aggravations. In particular, the presence of abnormal shadows in more than 3 lobes was significantly associated with radiological aggravations.

### Conclusions

In this study, the presence of extensive radiological findings and the absence of lung diseases other than MAC were predictors of radiological aggravations of treatment-naïve

sent to the e-mail address of the managing office of the Ethics Committee at ohashi.rinri@ext.toho-u.ac.jp.

**Funding:** There was no specific funding for this research project.

**Competing interests:** The authors have declared that no competing interests exist.

**Abbreviations:** ALB, serum albumin; CRP, serum C-reactive protein; CT, computed tomography; FC, fibrocavitary; MAC, *Mycobacterium avium* complex; NB, nodular/bronchiectatic; NTM, : nontuberculous mycobacterium; TP, serum total protein.

pulmonary MAC disease. In particular, the presence of abnormal shadows in more than 3 lobes was significantly associated with radiological aggravations.

## Introduction

Epidemiologic data suggest that the incidence and prevalence of nontuberculous mycobacterium (NTM) infections are increasing in many countries[1–6]. *Mycobacterium avium* complex (MAC), including *M. avium* and *M. intracellulare*, is the most common etiology of NTM[7,8]. Although the progressive improvements in diagnostic technology such as chest computed tomography (CT) and genetic sequencing suggest that host and microorganism factors[9–12], as well as environmental factors[12], might be involved, the exact reason for the increasing prevalence of MAC remains unknown.

The clinical outcomes of pulmonary MAC disease vary widely. Some patients respond well to standard treatment including clarithromycin, ethambutol, and rifampicin, whereas others show resistance to standard treatment with poor outcomes, and some other patients remain stable without any treatment[7,13]. Thus, it is critical to determine the predictors for the prognosis of patients with pulmonary MAC disease.

It has long been considered that the causative species are critical predictors, but the difference in the prognosis between *M. avium* and *M. intracellulare* infections remains uncertain [14,15]. Other factors that are currently considered to aggravate pulmonary MAC disease are the presence of fibrocavitary type on radiography, a positive acid-fast bacilli (AFB) smear of sputum samples, and a larger affected area[7,14,16–18]. However, there have been few studies of the factors that exacerbate pulmonary MAC disease without treatment, and they were generally judged based on the initiation of treatment as an indicator of aggravation. Thus, the aim of the present study was to clarify the significant predictors of radiological aggravations of pulmonary MAC disease using only the data of treatment-naïve patients.

## Methods

### Study population

From April 2011 to December 2018 at our institute, of the patients with suspected NTM, 568 NTM cases were newly identified by cultures. Of them, 295 were diagnosed with NTM disease according to the 2007 American Thoracic Society/Infectious Disease Society guideline[7]. Those with a past history of NTM disease and NTM other than MAC were excluded from the present study. Finally, 238 subjects were enrolled in the present study (Fig 1). The clinical findings of the subjects, including age, sex, past history of tuberculosis, laboratory data, and radiological findings, were obtained from their medical records and retrospectively evaluated. To identify the significant predictors and to exclude treatment bias, analysis of only 167 treatment-naïve subjects (patients who did not receive medications for MAC during the observation period, and their radiographic findings were evaluated at the time of diagnosis and 1 year later) was performed.

### Ethics approval and consent to participate

This research was conducted using information previously collected in the course of normal care (without the intention to use it for research at the time of collection). The need for written, informed patient consent was waived in view of the retrospective and observational nature

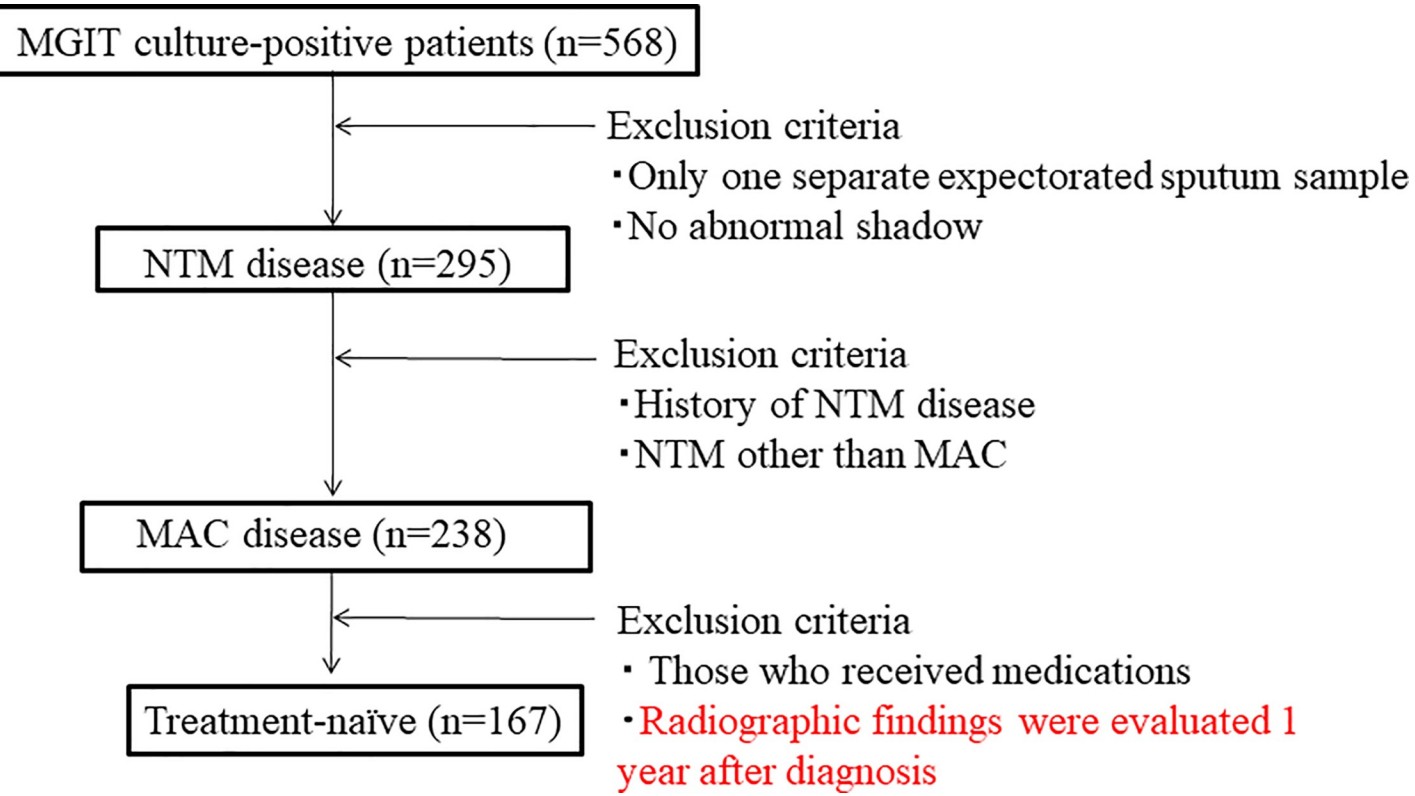

**Fig 1. Flow chart of patients diagnosed with pulmonary MAC disease between April 2011 and December 2018.** MGIT = mycobacterial growth indicator tube, NTM = nontuberculous mycobacterium, MAC = *Mycobacterium avium* complex.

of the study. This study received ethical approval from the Special Committee of Toho University Ohashi Medical Center, which is an ethics committee that reviews research on human subjects (project registration number H20004).

## Microbiological examination

AFB were cultured in a Mycobacteria Growth Indicator Tube (MGIT) from extracted sputum or bronchial washings obtained by bronchoscopy. The sputum samples were obtained on two or more occasions after the initial presentation. The diagnosis of MAC was confirmed when cultures were positive for AFB, and the cultured AFB was subsequently confirmed as MAC by PCR. The diagnosis of pulmonary MAC disease was established when MAC was identified in sputum at least twice or in bronchial washings[7].

## Radiological examination

According to a previous report[7], chest radiological findings were classified as fibrocavitary (FC) type or nodular/bronchiectatic (NB) type on high-resolution CT. Additionally, chest radiological findings at the time of initial diagnosis were scored as follows. The lung fields were divided into six zones based on anatomical structures, i.e. right upper, right middle, right lower, left upper, left lingular, and left lower. When any abnormal findings including cavities, bronchiectasis, small nodules, consolidations, atelectasis, and so on were found in a zone at the time of diagnosis, they were each counted as one point and summed up in the six zones (maximum 6 points). The subjects were further classified based on their radiological imaging

findings during the follow-up period into three categories: exacerbation, no change, or improvement. Each category was defined as follows: exacerbation, abnormal shadows increased; no change, abnormal shadows remained stable on the whole; and improvement, abnormal shadows decreased. The three categories were classified by five respiratory specialists in a blinded fashion.

### Patient management

When patients did not receive medications for MAC during the observation period, radiographic findings were evaluated at the time of the diagnosis and 1 year later. The patients who received medications for MAC during the observation period, not only those who began guideline-based therapy, but also those who discontinued medications, were excluded in the analysis of treatment-naïve subjects.

### Statistical analysis

The patients' characteristics are presented as medians (interquartile range). Numerical data are expressed as numbers (%). To identify factors related to pulmonary MAC disease in treatment-naïve patients, univariate and multivariate logistic regression analyses were used to estimate odds ratios (ORs) with 95% confidence intervals (CIs) for radiological aggravation. Additional analysis was added regarding the affected area that was significant as a factor aggravating pulmonary MAC disease. The sensitivity and specificity of the radiological aggravation prediction model were calculated for each score value. The performance of the radiological aggravation prediction model was evaluated using the receiver operating characteristic (ROC) curve by calculating the area under the ROC curve[19,20]. All analyses were performed using SPSS Statistical software (version 22.0; IBM Japan, Tokyo, Japan). *P* values < 0.05 were considered significant.

## Results

### Baseline characteristics of pulmonary MAC patients

During the study period, 238 patients with pulmonary MAC disease were enrolled, and their baseline characteristics are summarized in Table 1. The causative organisms included *M. avium* (189/238, 79.4%), *M. intracellulare* (36/238, 15.1%), and mixed infections (13/238, 5.5%). All patients were HIV-negative. Their median age was 76 (68–82) years, and 80% of patients were over 65 years of age. Female (68.1%), never-smoker (64.7%), and NB type (80.6%) were predominant in MAC patients. The median BMI was slightly low (19.0 kg/m$^2$). Medications for MAC were given to 62 (26%) patients during the observation period. The median number of abnormal lung zones was 3.

### Predictors of exacerbation in treatment-naïve pulmonary MAC subjects

To exclude the bias of treatment because the treatment period was not fixed, univariate analysis was performed using only the data of treatment-naïve subjects (Table 2).

The univariate analysis showed that no lung diseases other than MAC, more extensive radiological findings, and positive AFB smear were significantly associated with radiological aggravations. Multivariate analysis was performed with factors that showed significant differences on univariate analysis (no lung diseases other than MAC, more extensive radiological findings, and positive AFB smear) (Fig 2). The radiological factor (larger affected area) and absence of other lung disease were significantly associated with radiological aggravations. Additional analysis, ROC curve analysis, was performed regarding the affected area that was

**Table 1. Baseline characteristics of pulmonary MAC patients (n = 238).**

| | |
|---|---|
| Age (y) | 76 (68–82) |
| Sex (male/female) | 76 / 162 |
| Smoking history (current/past/never/unknown) | 2 / 72 / 154 / 10 |
| BMI (kg/m$^2$) | 19.0 (16.8–21.5) |
| TP (g/dL) | 7.4 (7.0–7.9) |
| ALB (g/dL) | 3.8 (3.35–4.1) |
| CRP (mg/dL) | 0.19 (0.04–0.99) |
| Previous tuberculosis, n (%) | 23 (9.7%) |
| Lung disease other than mycobacterial disease, n (%) | 96 (40.3%) |
| Smear/culture/BALF | 48 / 136 / 54 |
| Positive AFB smear, n (%) | 48 (20.2%) |
| *M. avium*/*M. intracellulare*/*M. avium+intracellulare* | 189 / 36 / 13 |
| FC type/NB type | 46 (19.3%) / 192 (80.7%) |
| MAC therapy during follow-up | 62 (26.1%) |
| Zone of radiological findings (n) | 3 (2–4) |

Data are expressed as medians (interquartile range) or numbers (%).

AFB = acid-fast bacilli, ALB = serum albumin, BALF = bronchoalveolar lavage fluid, CRP = serum C-reactive protein, FC = fibrocavitary, MAC = *Mycobacterium avium* complex, NB = nodular/bronchiectatic, TP = serum total protein.

significant as an aggravating factor of pulmonary MAC disease. In the zones of abnormal findings at the time of diagnosis, 7/74 (9.46%) of those with less than 2 zones affected had a radiological aggravation in one year, and 49/93 (52.7%) with more than 3 zones affected showed a radiological aggravation in one year (Fig 3). ROC curve analysis was performed to determine the threshold value when considering how many zones showing radiological findings was a risk. The ROC curve had an area under the curve of 0.765 for the radiological aggravation

**Table 2. Predictors of radiological aggravation in treatment-naïve pulmonary MAC subjects (n = 167).**

| | | Univariate analysis | |
|---|---|---|---|
| | | OR (95%CI) | *P*-value |
| Age (y) | 78(71–84) | 0.985 (0.957–1.013) | 0.297 |
| Sex (female) | 109(65.3%) | 1.728 (0.855–3.495) | 0.128 |
| Never-smoker(%) | 107(64.1%) | 1.946 (0.935–4.050) | 0.075 |
| BMI (kg/m$^2$) | 19.2(16.7–21.5) | 0.916 (0.818–1.026) | 0.129 |
| TP (g/dL) | 7.3(6.8–7.9) | 1.498 (0.937–2.396) | 0.092 |
| ALB (g/dL) | 3.6(3.3–4.1) | 1.456 (0.839–2.529) | 0.182 |
| CRP (mg/dL) | 0.23(0.04–1.12) | 0.925 (0.781–1.094) | 0.363 |
| Previous tuberculosis, n (%) | 20(12.0%) | 2.196 (0.855–5.639) | 0.102 |
| No lung disease other than MAC disease, n (%) | 80(47.9%) | 2.903 (1.468–5.740) | 0.002* |
| Positive AFB smear, n (%) | 32(19.2%) | 3.020 (1.358–6.715) | 0.007* |
| *M. intracellulare*(%) | 24(14.4%) | 0.922 (0.371–2.288) | 0.860 |
| FC type(%) | 24(14.4%) | 2.250 (0.938–5.400) | 0.069 |
| Zone of radiological findings, n (%) | 3(2–4) | 1.979 (1.512–2.591) | <0.001** |

See footnotes of Table 1 for expansions of abbreviations, OR = odds ratio

*: *P*<0.05

**: *P*<0.001

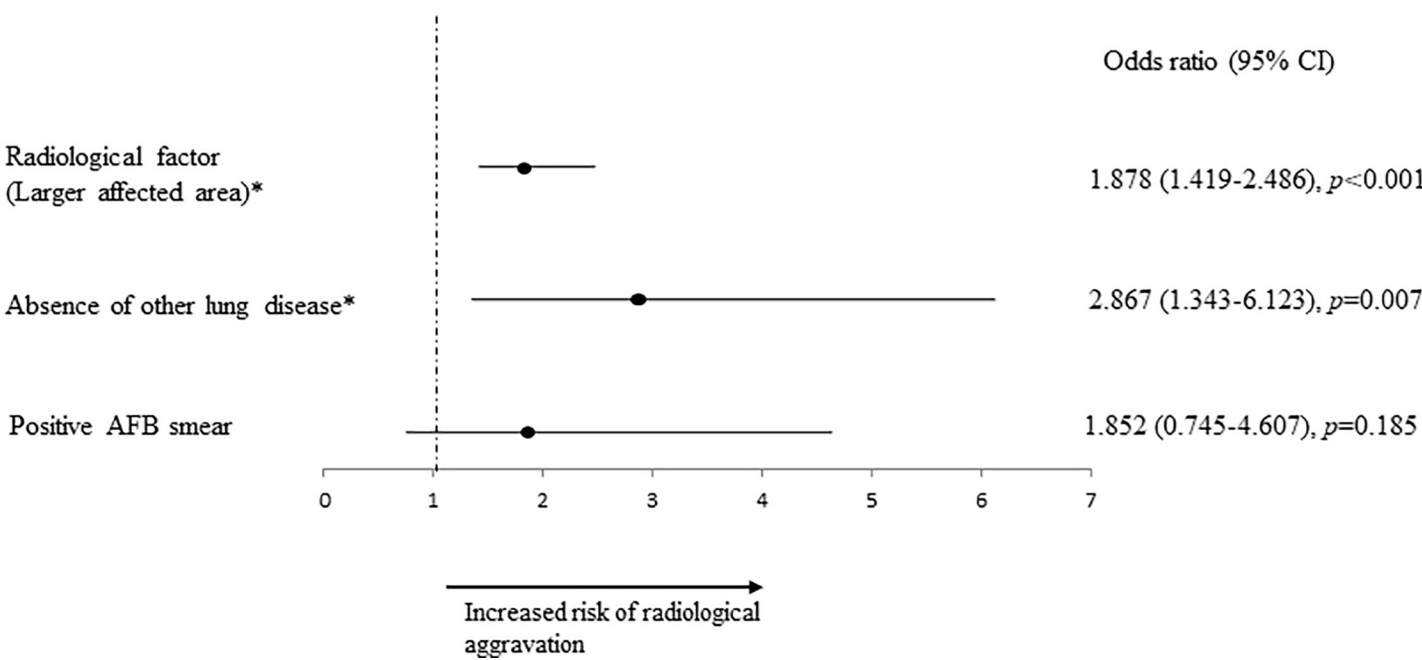

**Fig 2. ORs and associated 95% CIs for radiological aggravation of pulmonary MAC disease.** * Significant independent factors for radiological aggravation of pulmonary MAC disease.

prediction model (Fig 4). A threshold of 3 was identified as the optimal number of zones from the ROC curve, with a sensitivity of 87.5% and a specificity of 60.4%.

## Discussion

In the present study, *M. avium* was the predominant species, found in 79.4% of MAC patients. The rate of patients with *M. avium* infection was similar to other recent reports[21]. Similarly, there were more female than male patients in the present study. Generally, pulmonary MAC diseases develop more frequently in female than in male patients. In Japan, more women than men often work around water, and wet environmental exposure might be involved[12]. A recent biological study reported the role of estrogen in the development of pulmonary MAC disease, whereas the role of sex in disease susceptibility has yet to be determined[22]. Generally, pulmonary MAC diseases develop more frequently in thin and never smoker patients[23–25]. In the present study, similarly, patients had slightly low BMIs, and approximately two-thirds of pulmonary MAC disease patients were never smokers.

The aim of the present study was to clarify the significant predictors of radiological aggravations of pulmonary MAC disease using only the data of treatment-naïve patients. To date, disease progression of pulmonary MAC disease was defined as either requiring the start of treatment[7,14,17] or the presence of aggravation on radiological imaging[26,27]. In the present study, disease progression was defined as aggravation on radiological imaging. In some previous studies, the reason that the initiation of treatment was defined as an indicator of exacerbation was that MAC is indolent in nature, and thus, in many cases, radiological changes are difficult to evaluate on chest X-ray, detailed evaluation requires chest CT, and no radiological evaluation method for pulmonary MAC disease has been established globally. However, the timing of treatment may be biased by each doctor and each patient when using the initiation of treatment as evidence of exacerbation. For example, elderly patients tend to disagree with long-term medication, even if the doctor suspects deterioration and considers that they

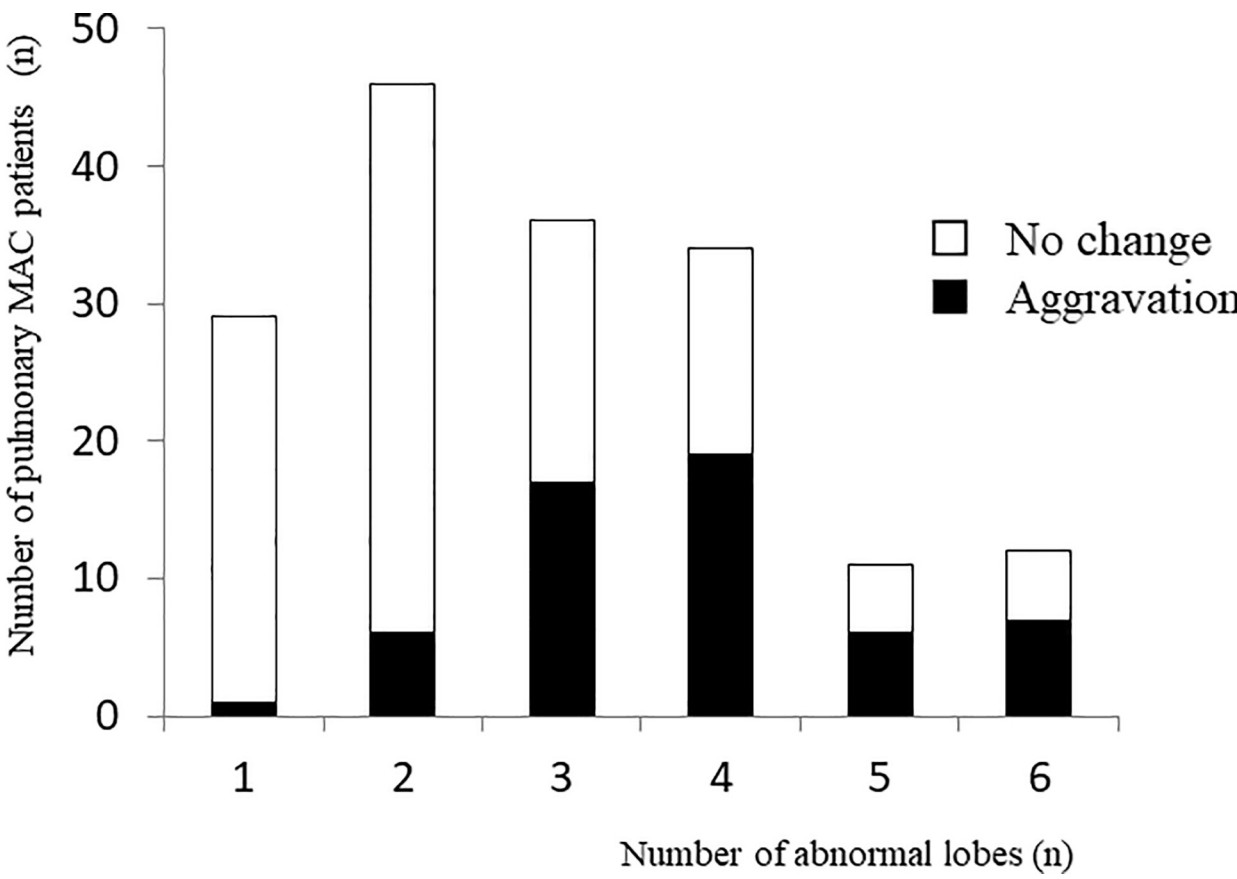

**Fig 3. Aggravation by number of lung lobes with abnormalities.** The black bar shows the number with radiological aggravation in each number of abnormal lobes, and the white bar shows that it has not changed. X-axis: Number of abnormal lobes, Y-axis: Number of pulmonary MAC patients.

should be treated. Fortunately, in most of the present cases, CT was performed in our hospital, and it was possible to examine the changes in radiological evaluations.

In the present study, radiological aggravation over one year was found in 56/167 (33.5%) of treatment-naïve subjects. Previous studies reported that about 20–40% and 50% of pulmonary MAC patients showed radiological aggravations after 5 and 10 years, respectively[16,26]. In the present study, the frequency of radiological aggravations was relatively high within only one year because of the absence of treatment. In the present study, the absence of other underlying lung diseases and the presence of more extensive radiological findings (larger affected area) in untreated patients were associated with radiological aggravations based on the probabilities of reactivation or dissemination of the infection. The present analysis indicated that the more extensive the radiological findings at initial diagnosis, the more likely a subsequent MAC aggravation becomes, which is in accordance with the findings of previous studies[17,18]. Previous studies reported that, in addition to extensive radiological findings, positive sputum AFB smear[7,17], FC type[7,16], and lower BMI[16,18] were aggravating factors. It is difficult to make a strict comparison between the current study and previous studies, because previous studies that treated patients are included, or they defined exacerbation as requiring treatment. In the present study, positive AFB smear, FC type, and lower BMI were not associated with radiological aggravations, but positive AFB smear (OR 3.020, 1.358–6.715) tended to be more common in patients with MAC disease aggravations.

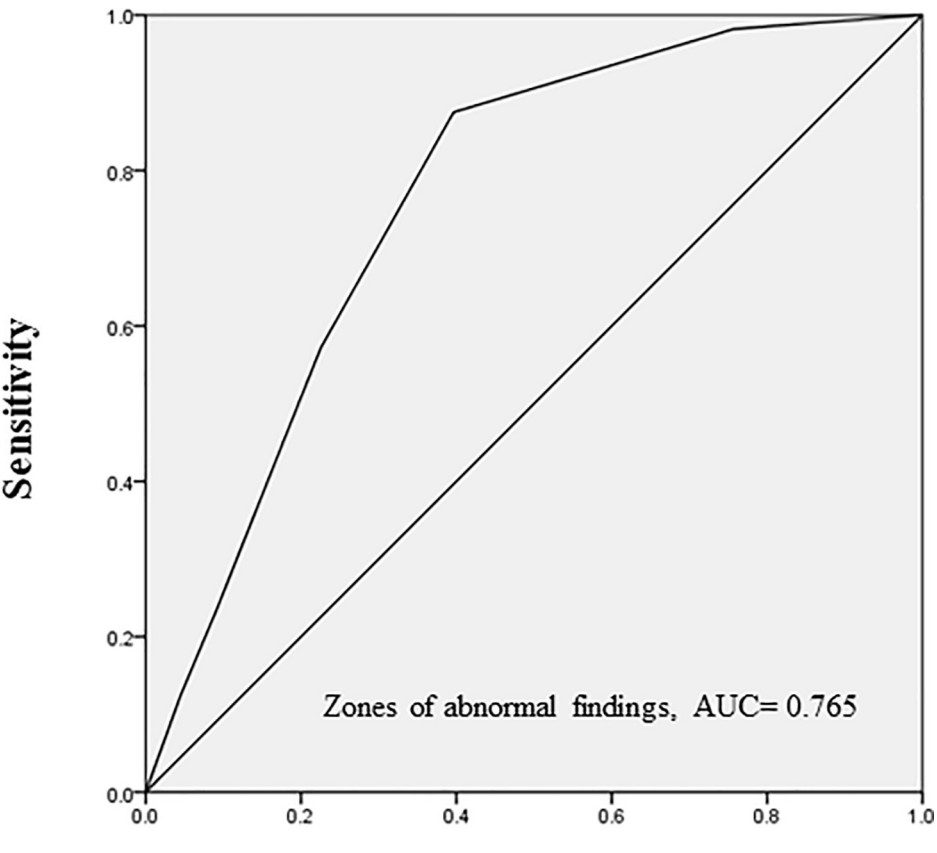

**Fig 4. The radiological aggravation prediction model: ROC curve.** ROC: receiver operating characteristic. AUC: area under the curve, Cut-off value: 3, sensitivity 87.5%, specificity 60.4%.

No studies examined the presence or absence of underlying lung diseases as an aggravating factor in pulmonary MAC disease. The present study demonstrated that the absence of other underlying lung diseases in untreated MAC patients was a significant aggravating factor of pulmonary MAC disease. Patients having other underlying lung diseases seemed to undergo radiological examinations more frequently than those without other underlying lung diseases. Thus, there may be more opportunities to identify the early phase of pulmonary MAC disease in those with underlying lung diseases. However, it cannot be ruled out that the diseases themselves, such as some kind of lung disease, and part of their treatment may be factors that suppress the progression of pulmonary MAC disease[28]. These will be our future research targets. Additionally, more extensive radiological findings were found to be an aggravating factor in the present analysis; therefore, a simple radiological scale assessment was performed with additional analysis by ROC curve analysis. Although some authors reported radiological scoring methods in pulmonary MAC disease[29,30], they were complicated and required much effort. Compared to these reports, the present scoring system had some limitations and merits. It simply counted the number of abnormal lesions, irrespective of their volume and characteristics. Nonetheless, it can be easily performed in actual clinical practice and can potentially predict the natural course of pulmonary MAC disease, as suggested by the present

report. With abnormal lesions in more than 3 zones, approximately half of the cases showed aggravation on imaging within one year, but with lesions in less than 2 zones, less than 10% showed aggravation. An ROC curve to determine the threshold value when considering how many zones of radiological findings are a risk identified 3 as the ideal threshold. Thus, treatment might be considered within one year when MAC disease involves 3 or more zones at the initial diagnosis.

## Limitations

Some limitations of the present study should be addressed. This study was limited by its retrospective nature without randomization, and it was a single-institution study, and as such, it is not representative of the national population. Additionally, this was a short-term study, and the number of MAC patients may have been underestimated since patients who were not diagnosed according to the 2007 American Thoracic Society/Infectious Disease Society guideline were excluded from the analyses. Therefore, factors with clinical significance in reality may have proven insignificant in the analyses with reduced statistical power.

## Conclusion

Women and never-smokers were predominant among patients with pulmonary MAC diseases. The critical factor for radiological aggravation of pulmonary MAC disease over a 1-year period is the presence of extensive abnormal shadows, especially the presence of abnormal shadows in ≥3 lobes in the lung. Thus, clinical attention should be focused on early diagnosis, because the presence of more extensive radiological findings (larger affected area) in untreated patients was associated with radiological aggravation.

## Author Contributions

**Data curation:** Chihiro Nakano, Takeshi Oshio, Kayo Watanabe, Kumiko Niitsuma, Chisato Imaizumi, Hiroto Matsuse.

**Formal analysis:** Norio Kodaka, Chihiro Nakano, Hiroto Matsuse.

**Investigation:** Norio Kodaka, Takeshi Oshio, Kayo Watanabe, Kumiko Niitsuma, Chisato Imaizumi, Hiroto Matsuse.

**Methodology:** Norio Kodaka, Hiroto Matsuse.

**Project administration:** Norio Kodaka.

**Supervision:** Hiroto Matsuse.

**Writing – original draft:** Norio Kodaka.

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
