## [Decision Letter · Decision Letter 0]

11 Dec 2019

PONE-D-19-29751

Predictors of exacerbations of pulmonary MAC disease

PLOS ONE

Dear Dr. Matsuse,

Thank you for submitting your manuscript to PLOS ONE. After careful consideration, we feel that it has merit but does not fully meet PLOS ONE’s publication criteria as it currently stands. Therefore, we invite you to submit a revised version of the manuscript that addresses the points raised during the review process.

ACADEMIC EDITOR: As you can see from the reviewers' comments, this manuscript needs extensive revision by adding required information as asked by the reviewers.  In addition, this article needs to be edited by a native English speaking, professional editorial service. Please incorporate all the edits/corrections on the manuscript and revise it accordingly before resubmission.

We would appreciate receiving your revised manuscript by Jan 24 2020 11:59PM. To enhance the reproducibility of your results, we recommend that if applicable you deposit your laboratory protocols in protocols.io, where a protocol can be assigned its own identifier (DOI) such that it can be cited independently in the future. For instructions see: http://journals.plos.org/plosone/s/submission-guidelines#loc-laboratory-protocols

We look forward to receiving your revised manuscript.

Kind regards,

Selvakumar Subbian, Ph.D.

Academic Editor

PLOS ONE

Journal Requirements:

**When submitting your revision, we need you to address these additional requirements:**

**Please ensure that your manuscript meets PLOS ONE's style requirements, including those for file naming. The PLOS ONE style templates can be found at http://www.plosone.org/attachments/PLOSOne_formatting_sample_main_body.pdf and http://www.plosone.org/attachments/PLOSOne_formatting_sample_title_authors_affiliations.pdf** Please provide additional details regarding participant consent. In the ethics statement in the Methods and online submission information, please ensure that you have specified (1) whether consent was informed and (2) what type you obtained (for instance, written or verbal). If your study included minors, state whether you obtained consent from parents or guardians. If the need for consent was waived by the ethics committee, please include this information.Please also clarify whether the 'Special Committee of Toho University Ohashi Medical Center' is an ethics committee that reviews research on human subjects. Your ethics statement must appear in the Methods section of your manuscript. If your ethics statement is written in any section besides the Methods, please move it to the Methods section and delete it from any other section. Please also ensure that your ethics statement is included in your manuscript, as the ethics section of your online submission will not be published alongside your manuscript. Thank you for stating in your Funding Statement: [The funders had no role in study design, data collection and analysis, decision to publish, or preparation of the manuscript.]Please provide an amended statement that declares *all* the funding or sources of support (whether external or internal to your organization) received during this study, as detailed online in our guide for authors at http://journals.plos.org/plosone/s/submit-now.  Please also include the statement “There was no additional external funding received for this study.” in your updated Funding Statement.Please include your amended Funding Statement within your cover letter. We will change the online submission form on your behalf.

Reviewers' comments:

Reviewer's Responses to Questions

**Comments to the Author**

1. Is the manuscript technically sound, and do the data support the conclusions?

Reviewer #1: No

Reviewer #2: No

2. Has the statistical analysis been performed appropriately and rigorously? 

Reviewer #1: No

Reviewer #2: No

3. Have the authors made all data underlying the findings in their manuscript fully available?

Reviewer #1: Yes

Reviewer #2: No

4. Is the manuscript presented in an intelligible fashion and written in standard English?

Reviewer #1: No

Reviewer #2: No

5. Review Comments to the Author

Reviewer #1: The data presented in the paper looks interesting but not novel. It is too preliminary to be accepted as a research article in your esteemed journal. It could be evaluated for a short note or communication after following changes are made.

Many of the parameter shows very high standard deviation. It would be more appropriate to give the interquartile range for all the parameters to understand the significance better. The conclusions are questionable in the current format.

The manuscript requires major revision in the written English language and cannot be accepted in the present format.

Reviewer #2: This is a retrospective study looking into factors associated with the exacerbation of pulmonary MAC infection. The statistical analysis mainly focused on the association between sex and smokers, as shown in Tables 2, 3 and 4. However, these analyses are indeed redundant and could be simplified into one table in Table 5 that looks into the main theme of this study, which is to identify the factors associated with the exacerbation of pulmonary MAC disease. An assumption could be made that the authors do not know the statistical method and taking such a redundant way. Besides, the multivariate analysis is not clear of what the authors are looking for, and the criteria for selecting the variates seem to be not justified making the results unreliable. Not only the statistical analyses are unreliable, but the results also are not new and tell the readers nothing additional to the clinical practice.

6. PLOS authors have the option to publish the peer review history of their article (what does this mean?). If published, this will include your full peer review and any attached files.

Reviewer #1: Yes: Dr. Radha Gopalaswamy

Reviewer #2: No

---

## [Author Response · Author response to Decision Letter 0]

9 Jan 2020

PONE-D-19-29751

Predictors of exacerbations of pulmonary MAC disease

PLOS ONE

Academic Editor

We wish to thank the academic editor for the comments.

Comment

1) Please provide additional details regarding participant consent. In the ethics statement in the Methods and online submission information, please ensure that you have specified (1) whether consent was informed and (2) what type you obtained (for instance, written or verbal). If your study included minors, state whether you obtained consent from parents or guardians. If the need for consent was waived by the ethics committee, please include this information.

Response

We added the following on Page 5, Lines 18-20: “The need for written informed patient consent was waived in view of the retrospective and observational nature of the study.”

Details about inclusion in MAC statistics are on the website for Toho University Medical Center Ohashi Hospital, and we obtained verbal informed consent for inclusion. The need for written informed patient consent was waived in view of the retrospective and observational nature of the study. As our study did not include minors, we did not obtain consent from parents or guardians.

Comment

Please also clarify whether the 'Special Committee of Toho University Ohashi Medical Center' is an ethics committee that reviews research on human subjects.

Response

We added the following on Page 5, Lines 22-23: “This committee is an ethics board that reviews research on human subjects.”

Comment

Your ethics statement must appear in the Methods section of your manuscript. If your ethics statement is written in any section besides the Methods, please move it to the Methods section and delete it from any other section. Please also ensure that your ethics statement is included in your manuscript, as the ethics section of your online submission will not be published alongside your manuscript.

Response

As you suggested, we moved the ethics statement to the Methods section and deleted it from the other section.

Comment

Reviewer #1: 

The data presented in the paper looks interesting but not novel. It is too preliminary to be accepted as a research article in your esteemed journal. It could be evaluated for a short note or communication after following changes are made.

Many of the parameter shows very high standard deviation. It would be more appropriate to give the interquartile range for all the parameters to understand the significance better. The conclusions are questionable in the current format.

The manuscript requires major revision in the written English language and cannot be accepted in the present format.

Response

We wish to thank the reviewer for the comments.

The new information provided by this study is that the most critical exacerbating factor of pulmonary MAC disease over 1 year is the presence of extensive abnormal shadows, especially the presence of abnormal shadows in ≥3 lobes in the lung. This information had never been reported by previous studies.

Our scoring system involves simply counting the number of abnormal lesions in the lung fields divided into six zones based on anatomical structures; thus, this system is far removed from previous detailed scoring systems, which are complicated and require much more effort. In contrast, our scoring system is simple and quick, so it can be easily performed in actual clinical practice.

As you suggested, we changed the date to interquartile ranges for all parameters to understand the significance better.

We also asked a native English-speaking medical editor to check the paper.

Comment

Reviewer #2:

This is a retrospective study looking into factors associated with the exacerbation of pulmonary MAC infection. The statistical analysis mainly focused on the association between sex and smokers, as shown in Tables 2, 3 and 4. However, these analyses are indeed redundant and could be simplified into one table in Table 5 that looks into the main theme of this study, which is to identify the factors associated with the exacerbation of pulmonary MAC disease. An assumption could be made that the authors do not know the statistical method and taking such a redundant way. Besides, the multivariate analysis is not clear of what the authors are looking for, and the criteria for selecting the variates seem to be not justified making the results unreliable. Not only the statistical analyses are

unreliable, but the results also are not new and tell the readers nothing additional to the clinical practice.

Response

We wish to thank the reviewer for the comments.

The number of patients with pulmonary MAC disease is increasing worldwide, especially among women and never-smokers. The roles of female hormones and adiponectin have been suggested, though the underlying mechanisms are yet to be determined. We did a simple analysis of this, and in the process, we found it was easier to detect MAC disease relatively early in smokers, due to the presence of other diseases. Therefore, we decided to perform additional analysis.

As you suggested, the presence of more extensive abnormal shadows has been previously shown to be associated with exacerbations of pulmonary MAC disease.

The new information provided by this study is that the most critical exacerbating factor of pulmonary MAC disease over 1 year is the presence of extensive abnormal shadows, especially the presence of abnormal shadows in ≥3 lobes in the lung. This information had never been reported by previous studies.

Our scoring system involves simply counting the number of abnormal lesions in the lung fields divided into six zones based on anatomical structures; thus, this system is far removed from previous detailed scoring systems, which are complicated and required much more effort. In contrast, our scoring system is simple and quick, so it can be easily performed in actual clinical practice. We therefore believe this paper merits publication.

Regarding statistics, some of the parameters showed high standard deviation. It would be more appropriate to give the interquartile range for all parameters to understand the significance better. Thus, we changed all parameters to include interquartile ranges.

---

## [Decision Letter · Decision Letter 1]

5 Feb 2020

PONE-D-19-29751R1

Predictors of exacerbations of pulmonary MAC disease

PLOS ONE

Dear Dr. Matsuse,

Thank you for submitting your manuscript to PLOS ONE. After careful consideration, we feel that it has merit but does not fully meet PLOS ONE’s publication criteria as it currently stands. Therefore, we invite you to submit a revised version of the manuscript that addresses the points raised during the review process.

ACADEMIC EDITOR: A major issue in this manuscript that has not been satisfactorily addressed is the statistical analysis of the retrospective data. Therefore, it is important that the authors explicitly explain the statistical analysis and the tools used etc., in the methods or results section (refer reviewer comment).  Also, it is important to discuss the limitations of this study. Include a paragraph in the discussion section that describes all type of limitations applicable to this retrospective analysis.

We would appreciate receiving your revised manuscript by Mar 21 2020 11:59PM. To enhance the reproducibility of your results, we recommend that if applicable you deposit your laboratory protocols in protocols.io, where a protocol can be assigned its own identifier (DOI) such that it can be cited independently in the future. For instructions see: http://journals.plos.org/plosone/s/submission-guidelines#loc-laboratory-protocols

We look forward to receiving your revised manuscript.

Kind regards,

Selvakumar Subbian, Ph.D.

Academic Editor

PLOS ONE

Reviewers' comments:

Reviewer's Responses to Questions

**Comments to the Author**

1. If the authors have adequately addressed your comments raised in a previous round of review and you feel that this manuscript is now acceptable for publication, you may indicate that here to bypass the “Comments to the Author” section, enter your conflict of interest statement in the “Confidential to Editor” section, and submit your "Accept" recommendation.

Reviewer #1: All comments have been addressed

Reviewer #2: (No Response)

2. Is the manuscript technically sound, and do the data support the conclusions?

Reviewer #1: Partly

Reviewer #2: Partly

3. Has the statistical analysis been performed appropriately and rigorously? 

Reviewer #1: No

Reviewer #2: No

4. Have the authors made all data underlying the findings in their manuscript fully available?

Reviewer #1: Yes

Reviewer #2: No

5. Is the manuscript presented in an intelligible fashion and written in standard English?

Reviewer #1: Yes

Reviewer #2: Yes

6. Review Comments to the Author

Reviewer #1: The manuscript titled “Predictors of exacerbation of pulmonary MAC disease” is written well and interesting to note the new scoring system for pulmonary involvement. The authors claim that “The new information provided by this study is that the most critical exacerbating factor of pulmonary MAC disease over 1 year is the presence of extensive abnormal shadows, especially the presence of abnormal shadows in ≥3 lobes in the lung. This information had never been reported by previous studies”. There is a certain bias in the age group who are all elders above 69. It does not represent the age distribution within a normal population. Besides, there are no representative images to explain the scoring system. The data on the co relation to sex and smoking status does not add any value to the idea mentioned in the title. Also, only 39 people have received MAC therapy. Although this is a very notable finding by the authors, in my opinion, it does not warrant its publication as a full-length manuscript but a brief communication.

Reviewer #2: I appreciate your response to my comments. Statistical methods are basically the same. As previously mentioned, there is nothing novel to our clinical practice. The author's comment indicates the presence of a scoring system, which is not mentioned in the manuscript.

7. PLOS authors have the option to publish the peer review history of their article (what does this mean?). If published, this will include your full peer review and any attached files.

Reviewer #1: Yes: Radha Gopalaswamy

Reviewer #2: No

---

## [Author Response · Author response to Decision Letter 1]

6 Mar 2020

PONE-D-19-29751

Predictors of exacerbations of pulmonary MAC disease

PLOS ONE

Academic Editor

We wish to thank the academic editor for the comments.

Comment

1) A major issue in this manuscript that has not been satisfactorily addressed is the statistical analysis of the retrospective data. Therefore, it is important that the authors explicitly explain the statistical analysis and the tools used etc., in the methods or results section (refer reviewer comment). Also, it is important to discuss the limitations of this study. Include a paragraph in the discussion section that describes all type of limitations applicable to this retrospective analysis.

Response

As you suggested, our statistical analysis of the retrospective data are difficult to understand, so we changed Table 5 as follows.

The odds ratio between the dependent factor (exacerbating factor) and the independent factor has been reduced to 1 or less.

Therefore, we changed “lung disease other than mycobacterial disease, n (%)” to “No lung disease other than MAC disease, n (%)” and corrected each numerical value in Table 5.

As mentioned, when we had consulted with experts about statistics, multivariate analysis with items that are similar was not considered statistically accurate. 

Therefore, multivariate analysis was performed using radiological factor (larger affected area), comorbidity (absence of other lung disease), sex (female sex), and smoking (never-smoker), and we created Figure 2 separately.

As regards limitations, we added the following on Page 11,Line 12, 

“Limitations Some limitations in our study should be addressed. This study is limited by retrospective nature without randomization, and single-institutional study. In addition, this study has been performed only for those with a definitive diagnosis of MAC disease. Suspected MAC disease, for example, the cases that bacteria were detected by only once with sputum , and the cases that have been treated with MAC diagnosis in the past, have been excluded. So, it is presumed that the number of cases was limited, the number of cases being treated was small, and the elderly were many.”

Comment

Reviewer #1: 

The manuscript titled “Predictors of exacerbation of pulmonary MAC disease” is written well and interesting to note the new scoring system for pulmonary involvement. The authors claim that “The new information provided by this study is that the most critical exacerbating factor of pulmonary MAC disease over 1 year is the presence of extensive abnormal shadows, especially the presence of abnormal shadows in ≥3 lobes in the lung. This information had never been reported by previous studies”. There is a certain bias in the age group who are all elders above 69. It does not represent the age distribution within a normal population. Besides, there are no representative images to explain the scoring system. The data on the co-relation to sex and smoking status does not add any value to the idea mentioned in the title. Also, only 39 people have received MAC therapy. Although this is a very notable finding by the authors, in my opinion, it does not warrant its publication as a full-length manuscript but a brief communication.

Response

We wish to thank the reviewer for the comments.

This study involved only those with a definitive diagnosis of MAC disease. Suspected cases of MAC disease, for example, the cases in which bacteria were detected only once in sputum and cases treated with a MAC diagnosis in the past, were excluded. Thus, it is presumed that the number of cases was limited, the number of cases being treated was small, and there were many elderly patients.

As you suggested, we added the following on Page11, Lines12, “Limitations Some limitations in our study should be addressed. This study is limited by retrospective nature without randomization, and single-institutional study. In addition, this study has been performed only for those with a definitive diagnosis of MAC disease. Suspected MAC disease, for example, the cases that bacteria were detected by only once with sputum and so on, the cases that have been treated with MAC diagnosis, have been excluded. So, it is presumed that the number of cases was limited, the number of cases being treated was small, and the elderly were many.”

As regards our scoring system, our scoring system involves just counting the number of abnormal lesions in the lung fields divided into six zones based on anatomical structures. Thus, we did not show the scoring system.

As mentioned, we deleted Table 5 that was not needed.

However, the number of patients with pulmonary MAC disease is increasing worldwide, especially among women and never-smokers. We did a simple analysis of it, and in the process, we noted that is easier to detect MAC disease relatively early in smokers, because of other diseases. Therefore, we decided to perform an additional analysis. 

Comment

Reviewer #2:

I appreciate your response to my comments. Statistical methods are basically the same. As previously mentioned, there is nothing novel to our clinical practice. The author's comment indicates the presence of a scoring system, which is not mentioned in the manuscript.

Response

We wish to thank the reviewer for the comments.

As mentioned, our statistical analysis of the retrospective data was difficult to understand, so we changed Table 5 as follows. The odds ratio between the dependent factor (exacerbating factor) and the independent factor was reduced to 1 or less. Therefore, we changed “lung disease other than mycobacterial disease, n (%)” to “No lung disease other than MAC disease, n (%)” and corrected each numerical value in Table 5.

As mentioned, when we had consulted with experts about statistics, multivariate analysis with the items that are similar was not considered statistically accurate. 

Therefore, multivariate analysis was performed using radiological factor (larger affected area), comorbidity (absence of other lung disease), sex (female sex), and smoking (never smoker), and we created Figure 2 separately.

I think, the new information provided by this study is that the most critical exacerbating factor of pulmonary MAC disease over 1 year is the presence of extensive abnormal shadows, especially the presence of abnormal shadows in ≥3 lobes in the lung, and tat female sex and never-smoker were not associated with exacerbations over 1 year.

---

## [Decision Letter · Decision Letter 2]

25 Mar 2020

PONE-D-19-29751R2

Predictors of exacerbations of pulmonary MAC disease

PLOS ONE

Dear Dr. Matsuse,

Thank you for submitting your manuscript to PLOS ONE. After careful consideration, we feel that it has merit but does not fully meet PLOS ONE’s publication criteria as it currently stands. Therefore, we invite you to submit a revised version of the manuscript that addresses the points raised during the review process.

ACADEMIC EDITOR: Unfortunately, this manuscript still lacks the scientific rigor and integrity. There are lots of issues in this manuscript including definition of "exacerbation" (refer reviewer#3 comments), limited number of samples, data interpretation and statistical analysis. As suggested by the reviewers, the authors should consider presenting the data of only treatment-naive patients in the main text. Also, summarize the limitations of this studies and discuss the discrepancy in the findings between this study and previously reported similar studies.

We would appreciate receiving your revised manuscript by May 09 2020 11:59PM. To enhance the reproducibility of your results, we recommend that if applicable you deposit your laboratory protocols in protocols.io, where a protocol can be assigned its own identifier (DOI) such that it can be cited independently in the future. For instructions see: http://journals.plos.org/plosone/s/submission-guidelines#loc-laboratory-protocols

We look forward to receiving your revised manuscript.

Kind regards,

Selvakumar Subbian, Ph.D.

Academic Editor

PLOS ONE

Reviewers' comments:

Reviewer's Responses to Questions

**Comments to the Author**

1. If the authors have adequately addressed your comments raised in a previous round of review and you feel that this manuscript is now acceptable for publication, you may indicate that here to bypass the “Comments to the Author” section, enter your conflict of interest statement in the “Confidential to Editor” section, and submit your "Accept" recommendation.

Reviewer #1: All comments have been addressed

Reviewer #3: (No Response)

2. Is the manuscript technically sound, and do the data support the conclusions?

Reviewer #1: Partly

Reviewer #3: Partly

3. Has the statistical analysis been performed appropriately and rigorously? 

Reviewer #1: No

Reviewer #3: No

4. Have the authors made all data underlying the findings in their manuscript fully available?

Reviewer #1: Yes

Reviewer #3: Yes

5. Is the manuscript presented in an intelligible fashion and written in standard English?

Reviewer #1: Yes

Reviewer #3: Yes

6. Review Comments to the Author

Reviewer #1: The paper is well written concise about a new scoring system and a pattern for MAC disease. Although interesting the study has limitations of its own and in my opinion does not warrant a publication as full length research article.

This can be an excellent short communication wherever suitable.

Reviewer #3: (No Response)

7. PLOS authors have the option to publish the peer review history of their article (what does this mean?). If published, this will include your full peer review and any attached files.

Reviewer #1: No

Reviewer #3: No

---

## [Author Response · Author response to Decision Letter 2]

7 May 2020

PONE-D-19-29751R2

Predictors of radiologic aggravations of pulmonary MAC disease

PLOS ONE

Academic Editor:

We wish to thank the academic editor for the comments.

Comment

There are lots of issues in this manuscript including definition of "exacerbation" (refer reviewer#3 comments), limited number of samples, data interpretation and statistical analysis. As suggested by the reviewers, the authors should consider presenting the data of only treatment-naive patients in the main text. Also, summarize the limitations of this studies and discuss the discrepancy in the findings between this study and previously reported similar studies.

Response

As you suggested, we now present the data only of treatment-naive patients in the main text. We also replaced “exacerbation” with “radiologic aggravation” and added the following on Page 10, Line 19, “In this study, disease progression was defined as aggravation on the radiological image.” 

As you pointed out, the study period was extended to December 2018 due to the small number of cases, and we also corrected the statistical analysis. 

We summarized the limitations of this study and discussed the discrepancy in the findings between this study and previously reported similar studies.

Reviewer

We wish to thank the reviewer for the comments.

Comment

The authors investigated the predictors of exacerbations of pulmonary MAC lung disease. Although the topic addressed in this manuscript is interesting, some of the interpretations underlying the authors' approach to be flawed. Additional description and further clarification is needed. 

Major comments.

1. The description of the results and the findings of this study were quite confusing. I think this is because the authors used two different definitions with regard to the definition of “exacerbation”. That is, “exacerbation” was defined as “the frequency of MAC therapy” in Table 2, Table 3, and Page 10 line 14-16. In contrast, “exacerbation” was defined as “radiologic aggravation” in Figure 2, Table 4, Page 9 line 1-12, and Page 10 line 12-14. Previous studies (Clinical Infectious Diseases 2017;65(6):927–34, Eur Respir J 2017; 49: 1600537, BMJ Open 2015;5:e008058, http://dx.doi.org/10.5588/ijtld.13.0792) adopted either, but not both, as a definition of exacerbation. 

Response

As suggested, we deleted Table 2 and Table 3, and we present the data only of treatment-naive patients in the main text. We also replaced “exacerbation” with “radiologic aggravation” and added the following on Page 10, Line 19, “In this study, disease progression was defined as aggravation on the radiological image.”

Comment

2. Moreover, in contrast to the authors’ description in Page 4 line 17-21, I think there is no previous studies reporting that sex and smoking status were related to the exacerbation (the frequency of MAC therapy) of MAC lung disease. I believe that, if there is any relationship, this may be due to the higher number of fibrocavitary types in men and smokers compared with women and non-smokers. Although radiologic types were not related to the exacerbation in this study, it was likely that the number of enrolled subjects were small to reveal clinical significance. Likewise, there is no reason that sex and smoking status might be related to the radiologic deterioration of MAC lung disease. Therefore, I think that Table2,3, and its relevant descriptions in the Results and Discussion sections are unnecessary because these results had little clinical implications. 

Response

As you suggested, there have been no previous studies showing that sex and smoking status were related to the exacerbation (the frequency of MAC therapy) of MAC lung disease.

Therefore, we changed the following on Page 4, Line 21, “In contrast to prevalence, whether sex and smoking status could be factors aggravating pulmonary MAC disease remains unknown.”

As you pointed out, the study period was extended to December 2018 due to the small number of cases, and we also corrected the statistical analysis.

Comment

3. In contrast to previous studies concerning the predictors of radiologic deterioration enrolled both the patients received treatment and treatment-naïve patients (Clinical Infectious Diseases 2017;65(6):927–34, BMJ Open 2015;5:e008058, http://dx.doi.org/10.5588/ijtld.13.0792), the authors performed their analysis of predictors of exacerbation using only the data of treatment-naïve patients. I think this is the novel finding of this study. I recommend that the authors focus solely on this part and revise the paper as a whole. 

Response

As you suggested, we focused solely on the data of treatment-naïve patients and revised the paper as a whole.

Comment

4. The term “exacerbation” is confusing. Because “exacerbation” is usually referred to the clinical situation that required treatment. I think “radiologic deterioration” or “radiologic aggravation” should be used instead of “exacerbation” throughout the manuscript. 

Response

As you suggested, we replaced “exacerbation” with “radiologic aggravation” throughout the manuscript.

Comment

5. Page 6, line 6: Reference is needed for this description. 

Response

We added a reference.

Comment

6. The authors should add the data about the number and the percentage of patients with aggravation, no change, and improvement. 

Response

As previously mentioned, we deleted Table 2 and Table 3.

Comment

7. Page 7, line 4-line 6

The analysis of radiographic findings in patients received treatment was not found in the manuscript. However, as I suggested above, I recommend authors to revise the manuscript excluding the patients underwent treatment. 

Response

As you suggested, we revised the manuscript excluding the patients who underwent treatment.

Comment

8. Statistical analysis

1) For the multivariate logistic analysis, the odds ratio (OR) is used. The hazard ratio is used for the Cox regression analysis. 

2) In addition, higher OR should be associated with an increased risk of radiologic deterioration. However, in Figure 2, lower OR was associated with an increased risk of radiologic deterioration. 

Response

As you suggested, we changed “HR” to “OR” and changed it so that higher OR was associated with an increased risk of radiologic aggravation. 

Comment

9. In Table 1, all data are expressed as median value. However, in the Results section, the authors described “average” BMI and “average” number of abnormal lung zone. The mean and median values should be clearly distinguished. Please note that variables with normal distribution should be expressed as mean +/- standard deviation.

Response

In the previous revision, it was pointed out that the mean ± standard deviation should be changed to the median value due to the small number of cases. As suggested, we clearly distinguished the mean and median values.

Comment

10. Author might want to add new Table according to the radiologic aggravation, instead of Table 4. 

Response

As suggested, we changed “exacerbation” to “radiologic aggravation.”

Comment

11. In Table 4, “no” lung disease other than MAC disease was a statistically significant predictor in univariate analysis. However, in the Result and Discussion section, it was described that “fewer” lung disease other than MAC was a significant factor. I think that “no” and “fewer” was totally different. 

Response

As suggested, we corrected it to “no lung disease other than MAC”.

Comment

12. In Table 4, it seems that “zone of radiologic findings”, involved lobes ≥2, and involved lobes ≥3 were variables that were highly correlated with one another. In the multivariate logistic analysis, if two or more variables are highly correlated with one another, it is hard to get good estimates of their distinct effects on some dependent variable. The authors should check multicollinearity in their statistical analysis. 

Response

As you suggested, if two or more variables are highly correlated with one another, it is hard to obtain good estimates of their distinct effects on a dependent variable.

Therefore, regarding the number of abnormal lesions, the sensitivity and specificity of the radiologic aggravation prediction model were calculated for each score value. We added Figure 3 and Figure 4. The performance of the radiologic aggravation prediction model was evaluated using the receiver operating characteristic (ROC) curve with the calculation of the area under the ROC curve.

Comment

13. Page 10, line 4-line 5

Such results cannot be found in the manuscript. 

Response

We deleted the sentence.

Minor comments.

1. Page 4, line 10-line 11: The authors might want to use “guideline-based therapy” instead of “standard therapy”.

Response

As you suggested, we used “guideline-based therapy” instead of “standard therapy”.

Comment

2. Page 6, line 19: clinical course � radiologic image? 

Response

We changed “clinical course” to “radiologic image”.

Comment

3. Page 8, line 1: M. avium – 100/128 (78.1%), M. intracellulare – 21/128 (16.4%), mixed infection – 7/128 (5.5%)

Response

We corrected it.

Comment

4. Page 8, line 5: BMI 19.1 – However, BMI was 19.0 in Table 1

Response

We corrected it.

Comment

5. Tables: positive MAC smear � positive AFB smear

Response

We replaced “positive MAC smear” with “positive AFB smear,” as suggested.

Comment

6. Page 10, line 18, Page 11, line 1: treatment “resistance” is an inappropriate word.

Response

We deleted the term.

---

## [Decision Letter · Decision Letter 3]

29 May 2020

PONE-D-19-29751R3

Predictors of radiologic aggravations of pulmonary MAC disease

PLOS ONE

Dear Dr. Matsuse,

Thank you for submitting your manuscript to PLOS ONE. After careful consideration, we feel that it has merit but does not fully meet PLOS ONE’s publication criteria as it currently stands. Therefore, we invite you to submit a revised version of the manuscript that addresses the points raised during the review process.

ACADEMIC EDITOR: The authors should pay serious attention in reviewing their manuscript for uniformity, language and order to improve the quality. Take into consideration all the reviewer comments and address them at the appropriate place in the manuscript.

We look forward to receiving your revised manuscript.

Kind regards,

Selvakumar Subbian, Ph.D.

Academic Editor

PLOS ONE

Reviewers' comments:

Reviewer's Responses to Questions

**Comments to the Author**

1. If the authors have adequately addressed your comments raised in a previous round of review and you feel that this manuscript is now acceptable for publication, you may indicate that here to bypass the “Comments to the Author” section, enter your conflict of interest statement in the “Confidential to Editor” section, and submit your "Accept" recommendation.

Reviewer #1: All comments have been addressed

Reviewer #3: All comments have been addressed

2. Is the manuscript technically sound, and do the data support the conclusions?

Reviewer #1: Partly

Reviewer #3: Yes

3. Has the statistical analysis been performed appropriately and rigorously? 

Reviewer #1: Yes

Reviewer #3: Yes

4. Have the authors made all data underlying the findings in their manuscript fully available?

Reviewer #1: Yes

Reviewer #3: Yes

5. Is the manuscript presented in an intelligible fashion and written in standard English?

Reviewer #1: No

Reviewer #3: Yes

6. Review Comments to the Author

Reviewer #1: The manuscript “Predictors of radiologic aggravations of pulmonary MAC disease” is a neat and simple analysis. However, discussion of the results obtained should be more precise and clearer to benefit the readers. Grammar and language can be redone more rigourously. Overall formatting needs to be uniform. References can be more updated ones

Revisions:

Page 4

Line 3 : Please update the references. The quoted ones are more than a decade old.

Page 8

Lines 13-17 can be simplified for easy understanding

Line 18 “Multivariate analysis was performed with items that showed significant differences on univariate analysis”. Please change the word “items” to a more scientific word.

Line 23 Please elaborate “additional analysis”

Page 9

Line 16 -18 “A recent biological study reported the role of oestrogen in the development of pulmonary MAC disease, whereas the role of sex in disease susceptibility has yet to be determined”

The quoted reference dates back to 2001. Please use an update reference or avoid the word recent.

Line 18-19 “Generally, pulmonary MAC diseases develop more frequently in thin patients”. Please provide appropriate reference.

Line 20 – Nutritional status was suggested by whom. Please be more scientific in discussion.

Line 22 – “Adiponectin might be involved”. This could be elaborated.

Page 10, 11 When you discuss the association of positive smear as an aggravating factorm it needs to be more precise and clearer. It is better to rewrite the univariate and multivariate analysis results more clearly.

Table 2: Please change the “See footnotes of Tables 1 and 2 for expansions of abbreviations “ to table 1 and not as table 1 and 2. Mention interquartile ranges and significant * p value for benefit of readers.

Figure 3: please clearly define the Y axis and explain the same in legend

Figure 4: Please explain the plot and mention the cutoff rather than just giving the title

Reviewer #3: The authors have responded well to my suggestions and revised the paper superbly. However, some major corrections and further clarification are still strongly needed.

Major comments.

1. Page 4, line 17-22

After major revision, the authors have revised their manuscript by deleting the findings and its relevant descriptions concerning the sex and smoking status (such as Table 2 and Table 3 in the previous manuscript). Therefore, I think that these sentences should be deleted (or at least modified) in the revised manuscript. Instead, authors should add the descriptions about the “radiologic aggravation”, such as the results of the previous studies and unknown findings so far.

2. Page 3, line 10

Keywords should be modified. Namely, “sex” and “smoking status” should be removed from the keywords. Instead, authors might want to add the other keywords related to the radiological aggravation.

3. Page 8, line 1, Page 8, line 7, Figure 1, Table 2

238 (the number of total patients with MAC lung disease) – 62 (the number of patients received treatment) = 176

However, in Figure 1 and Table 2, the number of treatment-naïve patients was 167. Please clarify this.

4. Discussion

1) The sentence and paragraph of Discussion section is too fragmentary (In particular, Page 10, line 5-line 14, Page 11, line 21-line 23). Authors should rewrite Discussion section in a more comprehensive way.

2) In addition, I think that many descriptions in the Discussion section are generally irrelevant (In particular, Page 9 line 11-page 10 line 4, Page 10 line 12-line 14) to the main findings of the present study.

3) Page 10, line 19-Page 11, line 4: These sentences are simple repetitions of those described in the Result section.

4) Page 11, line 4 “female sex, never-smoker”: In the revised manuscript, I think that there is no more reason to make an assumption that gender or smoking status are related to radiologic aggravation.

5) Page 11, line 9: these studies -> previous studies

6) Page 11, line 9-line 11: Please add reference for this sentence.

7) Page 11, line 12: the presence -> the presence or absence

8) Page 11, line 19-line 20: Please delete this sentence.

9) Page 12, line 2: limitations -> I think that the Authors’ scoring system has “merit” over the previous scoring system, rather than “limitation”.

10) Page 12, line 16-19: I do not believe that this is the limitation of the present study. Because the aim of the present study is to investigate the predictors of radiologic aggravation in “treatment-naïve” patients diagnosed with MAC lung disease “according to the ATS criteria”.

11) Page 12, line 19-20 “the number of cases being treated was small”: Authors have already excluded the patients who received treatment. Therefore, these patients were not included in the main analysis of the present study.

12) Page 12, line 24 “the most critical factor”: As figure 2 shows, “no lung disease other than MAC” has higher OR than radiologic involvement. Therefore, I do not understand why “more extensive radiological findings” was the “most” critical factor of radiologic aggravation.

Minor comments.

1. Page 2, line 14-15: the natural predictors of exacerbation -> the predictors of radiologic aggravation

2. Page 2, line 18: were common -> were predominant

3. Page 5, line 13: also -> please delete

4. Page 7, line 3-line 6 -> please consider to delete this sentence

5. Page 8, line 16-17, “whereas female sex ~ aggravations” -> please delete this sentence.

6. Table 1, smoking history: 2+76+154+10 = 242, not 238

7. Table 1, previous tuberculosis: 23/238 = 9.7%, not 8.8%

8. Table 2: see footnotes of Table 1 and Table 2 -> see footnote of Table 1

7. PLOS authors have the option to publish the peer review history of their article (what does this mean?). If published, this will include your full peer review and any attached files.

Reviewer #1: No

Reviewer #3: No

---

## [Author Response · Author response to Decision Letter 3]

1 Jul 2020

PONE-D-19-29751R3

Predictors of radiologic aggravations of pulmonary MAC disease

PLOS ONE

Academic Editor:

We wish to thank the academic editor for the comments.

Our previous study period was extended to December 2018 due to the small number of cases; thus, we changed the project registration number with the ethics committee from H16053 to H20004.

Reviewer #1: The manuscript “Predictors of radiologic aggravations of pulmonary MAC disease” is a neat and simple analysis. However, discussion of the results obtained should be more precise and clearer to benefit the readers. Grammar and language can be redone more rigourously. Overall formatting needs to be uniform. References can be more updated ones

We wish to thank the reviewer for the comments.

Comment

Page 4

Line 3 : Please update the references. The quoted ones are more than a decade old.

Response

We updated the references (#1-#5).

Comment

Page 8

Lines 13-17 can be simplified for easy understanding

Response

We changed the following sentences.

“The results indicated that no lung diseases other than MAC, more extensive radiological findings, and positive AFB smear were significantly associated with radiologic aggravations, whereas female sex and never-smoker were not found to be significantly associated with radiologic aggravations.”

→”The univariate analysis showed that no lung diseases other than MAC, more extensive radiological findings, and positive AFB smear were significantly associated with radiological aggravations.”

Comment

Line 18 “Multivariate analysis was performed with items that showed significant differences on univariate analysis”. Please change the word “items” to a more scientific word.

Response

As you mentioned, we replaced “items” with “factors”.

Comment

Line 23 Please elaborate “additional analysis”

Response

We added “ ROC curve analysis” after additional analysis.

Comment

Page 9

Line 16 -18 “A recent biological study reported the role of oestrogen in the development of pulmonary MAC disease, whereas the role of sex in disease susceptibility has yet to be determined”

The quoted reference dates back to 2001. Please use an update reference or avoid the word recent.

Response

As you suggested, we updated the references (#22).

Comment

Line 18-19 “Generally, pulmonary MAC diseases develop more frequently in thin patients”. Please provide appropriate reference.

Response

We added the references (#23-25 ).

Comment

Line 20 – Nutritional status was suggested by whom. Please be more scientific in discussion.

Response

It was also pointed out that the nutritional status and so on, as an onset factor, has little relationship to the main findings of present study. Thus, we deleted the sentence about the onset factor appropriately.

Comment

Line 22 – “Adiponectin might be involved”. This could be elaborated.

Response

We think this is true, but, as described previously, we deleted the sentence about the onset factor.

Comment

Page 10, 11 When you discuss the association of positive smear as an aggravating factorm it needs to be more precise and clearer. It is better to rewrite the univariate and multivariate analysis results more clearly.

Response

To make the univariate and multivariate analysis results easier to understand, we added

the following on Page 8, Lines 16-17, “(no lung diseases other than MAC, more extensive radiological findings, and positive AFB smear)”

Comment

Table 2: Please change the “See footnotes of Tables 1 and 2 for expansions of abbreviations “ to table 1 and not as table 1 and 2. Mention interquartile ranges and significant * p value for benefit of readers.

Response

We replaced “Table 1 and Table 2” with “Table 1”.

We added interquartile ranges and significant * p values.

Comment

Figure 3: please clearly define the Y axis and explain the same in legend

Response

We clearly defined the Y axis and provided an explanation in the legend.

Comment

Figure 4: Please explain the plot and mention the cutoff rather than just giving the title

Response

We added “zones of abnormal findings, AUC= 0.765”in Figure 4 and the cutoff value in the Figure legend.

Reviewer #3: The authors have responded well to my suggestions and revised the paper superbly. However, some major corrections and further clarification are still strongly needed.

We wish to thank the reviewer for the comments.

Major comments.

Comment

Page 4, line 17-22

After major revision, the authors have revised their manuscript by deleting the findings and its relevant descriptions concerning the sex and smoking status (such as Table 2 and Table 3 in the previous manuscript). Therefore, I think that these sentences should be deleted (or at least modified) in the revised manuscript. Instead, authors should add the descriptions about the “radiologic aggravation”, such as the results of the previous studies and unknown findings so far.

Response

As you mentioned, we deleted the descriptions concerning sex and smoking status and changed to the following on Page 4, Lines 16-23, “Other factors that are currently considered to aggravate pulmonary MAC disease are the presence of fibrocavitary type on radiography, a positive acid-fast bacilli (AFB) smear of sputum samples, and a larger affected area.7,14,16-18 However, there have been few studies of the factors that exacerbate pulmonary MAC disease without treatment, and they were generally judged based on the initiation of treatment as an indicator of aggravation. Thus, the aim of the present study was to clarify the significant predictors of radiological aggravations of pulmonary MAC disease using only the data of treatment-naïve patients.”

Comment

2. Page 3, line 10

Keywords should be modified. Namely, “sex” and “smoking status” should be removed from the keywords. Instead, authors might want to add the other keywords related to the radiological aggravation.

Response

We added “radiological aggravation” instead of “sex” and “smoking status” as a keyword.

Comment

3. Page 8, line 1, Page 8, line 7, Figure 1, Table 2

238 (the number of total patients with MAC lung disease) – 62 (the number of patients received treatment) = 176

However, in Figure 1 and Table 2, the number of treatment-naïve patients was 167. Please clarify this.

Response

We excluded 9 patients who could not be evaluated by chest CT one year after MAC diagnosis. Thus, we added the following on Page 5, Line 13 and the exclusion criteria of Figure 1 “radiographic findings were evaluated at the time of diagnosis and 1 year later.”

Comment

4. Discussion

1) The sentence and paragraph of Discussion section is too fragmentary (In particular, Page 10, line 5-line 14, Page 11, line 21-line 23). Authors should rewrite Discussion section in a more comprehensive way.

Response

As you mentioned, we changed the sentences, and rewrote the following on Page 9, line 19-Page 10, line 13, “The aim of the present study was to clarify the significant predictors of radiological aggravations of pulmonary MAC disease using only the data of treatment-naïve patients. To date, disease progression of pulmonary MAC disease was defined as either requiring the start of treatment7,14,17 or the presence of aggravation on radiological imaging26,27. In the present study, disease progression was defined as aggravation on radiological imaging. In some previous studies, the reason that the initiation of treatment was defined as an indicator of exacerbation was that MAC is indolent in nature, and thus, in many cases, radiological changes are difficult to evaluate on chest X-ray, detailed evaluation requires chest CT, and no radiological evaluation method for pulmonary MAC disease has been established globally. However, the timing of treatment may be biased by each doctor and each patient when using the initiation of treatment as evidence of exacerbation. For example, elderly patients tend to disagree with long-term medication, even if the doctor suspects deterioration and considers that they should be treated. Fortunately, in most of the present cases, CT was performed in our hospital, and it was possible to examine the changes in radiological evaluations. In the present study, radiological aggravation over one year was found in 56/167 (33.5%) of treatment-naïve subjects. Previous studies reported that about 20-40% and 50% of pulmonary MAC patients showed radiological aggravations after 5 and 10 years, respectively.16,26. In the present study, the frequency of radiological aggravations was relatively high within only one year because of the absence of treatment.”

Comment

2) In addition, I think that many descriptions in the Discussion section are generally irrelevant (In particular, Page 9 line 11-page 10 line 4, Page 10 line 12-line 14) to the main findings of the present study.

Response

As you mentioned, we think that page 9 line 11-page 10 line 4 is irrelevant to the main findings of the present study. We deleted page 10 lines 12-14, as mentioned. However, this is a summary statistic of MAC pulmonary disease, and in order to prove that there is no significant difference from the general pulmonary MAC disease reports, it has been shortened significantly. 

Comment

3) Page 10, line 19-Page 11, line 4: These sentences are simple repetitions of those described in the Result section.

Response

As you mentioned, we deleted these sentences and changed to the following on Page 10, Line 19-Page 11, line12, “Previous studies reported that, in addition to extensive radiological findings, positive sputum AFB7,17 smear, FC type7,16, and lower BMI16,18 were aggravating factors. It is difficult to make a strict comparison between the current study and previous studies, because previous studies that treated patients are included, or they defined exacerbation as requiring treatment. In the present study, positive AFB smear, FC type, and lower BMI were not associated with radiological aggravations, but positive AFB smear (OR 3.020, 1.358-6.715) tended to be more common in patients with MAC disease aggravations.No studies examined the presence or absence of underlying lung diseases as an aggravating factor in pulmonary MAC disease. The present study demonstrated that the absence of other underlying lung diseases in untreated MAC patients was a significant aggravating factor of pulmonary MAC disease. Patients having other underlying lung diseases seemed to undergo radiological examinations more frequently than those without other underlying lung diseases. Thus, there may be more opportunities to identify the early phase of pulmonary MAC disease in those with underlying lung diseases. However, it cannot be ruled out that the diseases themselves, such as some kind of lung disease, and part of their treatment may be factors that suppress the progression of pulmonary MAC disease.28 These will be our future research targets.”

Comment

4) Page 11, line 4 “female sex, never-smoker”: In the revised manuscript, I think that there is no more reason to make an assumption that gender or smoking status are related to radiologic aggravation.

Response

Based on your comment, we deleted these sentences.

Comment

5) Page 11, line 9: these studies -> previous studies

Response

We replaced “these studies” with “previous studies”, as suggested.

Comment

6) Page 11, line 9-line 11: Please add reference for this sentence.

Response

We added a reference.

Comment

7) Page 11, line 12: the presence -> the presence or absence

Response

We replaced “the presence” with “the presence or absence”, as suggested.

Comment

8) Page 11, line 19-line 20: Please delete this sentence.

Response

We deleted this sentence, as suggested.

Comment

9) Page 12, line 2: limitations -> I think that the Authors’ scoring system has “merit” over the previous scoring system, rather than “limitation”.

Response

As you mentioned, we replaced “several limitations” with “some limitations and merits.”

Comment

10) Page 12, line 16-19: I do not believe that this is the limitation of the present study. Because the aim of the present study is to investigate the predictors of radiologic aggravation in “treatment-naïve” patients diagnosed with MAC lung disease “according to the ATS criteria”.

Response

Thank you for your comment. As suggested, we changed the limitations. “Some limitations of the present study should be addressed. This study was limited by its retrospective nature without randomization, and it was a single-institution study, and as such, it is not representative of the national population.Additionally, this was a short-term study, and the number of MAC patients may have been underestimated since patients who were not diagnosed according to the 2007 American Thoracic Society/Infectious Disease Society guideline were excluded from the analyses. Therefore, factors with clinical significance in reality may have proven insignificant in the analyses with reduced statistical power.”

Comment

11) Page 12, line 19-20 “the number of cases being treated was small”: Authors have already excluded the patients who received treatment. Therefore, these patients were not included in the main analysis of the present study.

Response

We deleted the sentences, and made the changes mentioned above.

Comment

12) Page 12, line 24 “the most critical factor”: As figure 2 shows, “no lung disease other than MAC” has higher OR than radiologic involvement. Therefore, I do not understand why “more extensive radiological findings” was the “most” critical factor of radiologic aggravation.

Response

Based on this comment, we deleted “most”.

Comment

Minor comments.

1. Page 2, line 14-15: the natural predictors of exacerbation -> the predictors of radiologic aggravation

Response

We corrected it as suggested.

Comment

2. Page 2, line 18: were common -> were predominant

Response

We changed it as suggested.

Comment

3. Page 5, line 13: also -> please delete

Response

We deleted it as suggested.

Comment

4. Page 7, line 3-line 6 -> please consider to delete this sentence

Response

We deleted this sentence as suggested.

Comment

5. Page 8, line 16-17, “whereas female sex ~ aggravations” -> please delete this sentence.

Response

We deleted this sentence as suggested.

Comment

6. Table 1, smoking history: 2+76+154+10 = 242, not 238

Response

We corrected it.

Comment

7. Table 1, previous tuberculosis: 23/238 = 9.7%, not 8.8%

Response

We corrected it.

Comment

8. Table 2: see footnotes of Table 1 and Table 2 -> see footnote of Table 1

Response

We corrected it.

---

## [Decision Letter · Decision Letter 4]

21 Jul 2020

Predictors of radiologic aggravations of pulmonary MAC disease

PONE-D-19-29751R4

Dear Dr. Matsuse,

We’re pleased to inform you that your manuscript has been judged scientifically suitable for publication and will be formally accepted for publication once it meets all outstanding technical requirements.

Kind regards,

Selvakumar Subbian, Ph.D.

Academic Editor

PLOS ONE

Additional Editor Comments (optional):

Reviewers' comments:

Reviewer's Responses to Questions

**Comments to the Author**

1. If the authors have adequately addressed your comments raised in a previous round of review and you feel that this manuscript is now acceptable for publication, you may indicate that here to bypass the “Comments to the Author” section, enter your conflict of interest statement in the “Confidential to Editor” section, and submit your "Accept" recommendation.

Reviewer #1: All comments have been addressed

2. Is the manuscript technically sound, and do the data support the conclusions?

Reviewer #1: Yes

3. Has the statistical analysis been performed appropriately and rigorously? 

Reviewer #1: Yes

4. Have the authors made all data underlying the findings in their manuscript fully available?

Reviewer #1: Yes

5. Is the manuscript presented in an intelligible fashion and written in standard English?

Reviewer #1: Yes

6. Review Comments to the Author

Reviewer #1: (No Response)

7. PLOS authors have the option to publish the peer review history of their article (what does this mean?). If published, this will include your full peer review and any attached files.

Reviewer #1: No

---

## [Editor Report · Acceptance letter]

27 Jul 2020

PONE-D-19-29751R4 

Predictors of radiological aggravations of pulmonary MAC disease 

Dear Dr. Matsuse:

I'm pleased to inform you that your manuscript has been deemed suitable for publication in PLOS ONE. Congratulations! Your manuscript is now with our production department. 

Kind regards, 

on behalf of

Dr. Selvakumar Subbian 

Academic Editor

PLOS ONE